# From Search to Decision: A Framework for Adversarially Robust Approximate Nearest Neighbor Search

## Abstract

We design robust Approximate Nearest Neighbor (ANN) algorithms for a setting where an adversary controls both the dataset and $Q$ adaptive queries.

Our primary contribution is a general framework that reduces search problems to a corresponding robust decision problem via a binary search tree construction. Given an oblivious decider, we robustify it by applying the Differential Privacy framework of Hassidim, Kaplan, Mansour, Matias, and Stemmer (JACM 2022), enhanced by privacy amplification via subsampling. For ANN specifically, the main challenge is designing the oblivious decider itself. To that end, we propose a sampling-based Locality-Sensitive Hashing (LSH) approach, inspired by the work of Aumüller, Har-Peled, Mahabadi, Pagh, and Silvestri (TODS 2022) on fair ANN. This method is made efficient against worst-case data distributions via a novel concentric LSH construction, which also yields an improved algorithm for the exact fair ANN problem. The result is a simple, general, and efficient algorithm for all but a narrow class of degenerate datasets.

For the low-dimensional regime ($d = O(\sqrt{Q})$), we complement our general framework with specialized algorithms that provide a powerful "for-all" guarantee: correctness on every possible query with high probability. We propose novel metric covering constructions to simplify and improve prior approaches, enhancing performance for ANN in both Hamming and $\ell_p$ spaces.

## 1 Introduction

Randomness is a crucial tool in algorithm design, enabling resource-efficient solutions by circumventing the worst-case scenarios that plague deterministic approaches [45]. The classical analysis of such algorithms assumes an *oblivious* setting, where data updates and queries are fixed beforehand. However, this assumption breaks down in the face of an *adaptive adversary*, who can issue queries based on the algorithm's previous outputs. These outputs can leak information about the algorithm's internal randomness, allowing an adversary to construct query sequences that maliciously break the algorithm's performance guarantees [37, 34].

Significant progress has been made in designing adversarially robust algorithms for *estimation problems*, where the output is a single value [44, 38, 20, 8, 15, 54, 25]. A common defense involves sanitizing the output, for example, by rounding or adding noise, often borrowing techniques from differential privacy to ensure the output reveals little about the algorithm's internal state [38, 8, 14]. However, these techniques do not directly apply to *search problems*. In a search problem, the algorithm must return a specific element from a given dataset. Outputting a raw data point can leak substantial information, and there is no obvious way to add noise or otherwise obscure the output without violating the problem's core constraint of returning a valid dataset element.

Submitted to 39th Conference on Neural Information Processing Systems (NeurIPS 2025). Do not distribute.

Perhaps the most fundamental search problem is *Approximate Nearest Neighbor (ANN) Search*, which has numerous applications ranging from data compression and robotics to DNA sequencing and anomaly detection [47, 42, 39, 52, 51, 17]. The goal is to build a data structure that, for any query point, quickly finds a data point that is nearly the closest. Achieving the desired trade-off of sublinear query time and near-linear space has largely been possible only through randomization. The most prominent family of randomized algorithms for ANN is based on *Locality-Sensitive Hashing (LSH)*, which has been the subject of a long and fruitful line of research in the oblivious setting [33, 41, 1, 3, 7, 4, 2, 5, 40, 18].

The vulnerability of these classical randomized structures was recently highlighted by Kapralov, Makarov, and Sohler [43], who demonstrated an attack on standard LSH data structures. They showed that an adaptive adversary can use a polylogarithmic number of queries to learn enough about the internal LSH hashes to force the algorithm to fail. This attack leverages the very phenomenon that the algorithm's outputs (specific data points) reveal information about its random choices. Inspired by their work, which relies on certain structural properties of the dataset (e.g., an "isolated" point), we consider a powerful adversarial model where the **adversary chooses both the dataset and the sequence of queries**, posing a stringent test for robustness. We study the following question:

*Can search problems like ANN be solved efficiently in the face of adversarial queries?*

## 1.1 Our Results and Techniques

We provide an extensive study of adversarially robust algorithms for ANN. If the adversary provides $Q$ adaptively chosen queries to the algorithm and the metric space $\mathcal{M}$ is $d$-dimensional, we examine two regimes:

### 1.1.1 $d = \omega(\sqrt{Q})$: Reduction to Decision Problems and Differential Privacy

Our main technical contribution is a meta-algorithm that solves search problems robustly by reducing them to an underlying *robust decision problem*. For ANN, this decision problem is simply: given a query $q$, does any "close" neighbor exist within a distance of $r$? The framework first robustifies an oblivious algorithm for the decision problem by applying techniques from Differential Privacy [38]. With a robust decider $\mathcal{A}$ in hand, we solve the search problem by building a binary search tree of $\mathcal{A}$-instances on different parts of the input dataset.

A key challenge, however, is designing the initial oblivious decider for ANN. Standard LSH algorithms are unsuitable because of a critical ambiguity: for a query $q$, if the nearest neighbor lies in the annulus between distance $r$ and $cr$, LSH may either return that neighbor or report that no point within distance $r$ exists. Since both outcomes are valid for *approximate* search, LSH cannot be used to reliably decide if the $r$-ball around $q$ is empty. To overcome this, we draw inspiration from work on *fair ANN* [10], which aims to sample uniformly from all valid answers. We develop a sampling-based LSH algorithm that resolves this ambiguity, creating an efficient oblivious decider.

The performance of this sampling decider, however, depends on the density ratio of points between the $cr$-ball and the $r$-ball for a query $q$. An adversary can craft a dataset where this ratio is large, severely degrading performance. To mitigate this, we introduce a *concentric annuli construction*. We partition the $(r, cr)$-annulus into several smaller, concentric sub-annuli and apply our sampling LSH within each one. A simple counting argument guarantees that at least one of these sub-annuli must have a low point-density ratio, allowing the decider to terminate efficiently.

One limitation of our accelerated method is the existence of *degenerate* datasets (see Figure 1), which an adversary could construct to force a worst-case running time. We formalize this condition in Assumption 1 and argue that it corresponds to a contrived data distribution unlikely to appear in practice. For all non-degenerate datasets, our method remains efficient:

**Theorem 1.1.** *Let $K \geq 2$ be a parameter such that the input dataset $S$ is not $K$-degenerate under Assumption 1. Then, there exists an adversarially robust ANN algorithm using $\widetilde{O}(\sqrt{Q} \cdot n^{1+\rho+1/K})$ bits of space and $\widetilde{O}(dn^{\rho+1/K})$ time per query.*

Our method further provides an improved algorithm for the problem of sampling a near-neighbor uniformly at random (*fair ANN*), which we present in Appendix E:

**Theorem 1.2.** *Given Assumption 1, there exists an algorithm that on query $q \in \mathcal{M}$ outputs a point in $B_S(q, r')$ uniformly at random, where $r' \in [r, cr]$, and $c, r$ are the LSH parameters. If such points do not exist, the algorithm outputs "$\perp$" with high probability. The algorithm uses $O(n^{1+\frac{1}{K}+\rho} \cdot \log n)$ bits of space overall and $O(dn^{\rho+1/K} \log n)$ time per query.*

### 1.1.2 $d = O(\sqrt{Q})$: *For-all* Algorithms

For low-dimensional metric spaces, we develop algorithms for ANN that provide a powerful *for-all guarantee*: with high probability, the data structure correctly answers *every possible* query $q \in \mathcal{M}$. Our approach builds on a discretization technique applied to an LSH data structure, a paradigm explored in prior work [23, 24]. We refine this line of research by introducing a novel, simpler metric covering construction, improving the space complexity by a logarithmic factor, and using sampling to improve the time complexity by a factor of $d$.

**Theorem 1.3.** *For the $(c, r)$-ANN problem in the $d$-dimensional Hamming Hypercube, there exists an algorithm that correctly answers all possible queries with at least $0.99$ probability. The space complexity is $O(d \cdot n^{1+\rho+o(1)})$ and query time is $\widetilde{O}(d \cdot n^\rho)$, where $\rho = \frac{1}{2c-1}$.*

**Theorem 1.4.** *For the $(c, r)$-ANN problem in the $(\mathbb{R}^d, \ell_p)$ metric space, there exists an algorithm that correctly answers all possible queries with high probability. The pre-processing time and space are $\widetilde{O}(nT)$ and the query time is $\widetilde{O}(T/d)$, where:*

$$T = O\left(d \cdot n^{\rho'} \left(d \log d + \log n\right)\right) \quad and \quad \rho' = \frac{(10+c)^2}{161c^2 - 20c - 100}. \tag{1}$$

**Remark (The Price of For-All Algorithms).** *Despite their remarkable guarantees, for-all algorithms have significant drawbacks. Their space complexity scales by a factor of $d$, making them intractable for high-dimensional metric spaces. This is a direct consequence of the large number of hash functions required to ensure a tiny probability of error for any query. Furthermore, these algorithms lack the generality of their adaptive counterparts; they are metric-space dependent and must be tailored to the specific metric space being used.*

## 2 Preliminaries

**Definition 2.1 (Metric Balls).** *Consider a metric space $\mathcal{M} = (M, ||\cdot||)$, and let $S \subset \mathcal{M}$. We define a **metric ball** $B_S(x, r)$ on $S$ of radius $r$ centered at $x \in M$ as:*

$$B_S(x, r) := \{p \in S \mid ||x - p|| \leq r\}$$

*We often write $n(x, r) := |B_S(x, r)|$.*

In the *Nearest Neighbors* problem, we seek to find a point in our input dataset that minimizes the distance to some query point. Randomized algorithms are better suited to tackle the *approximate* version of the problem, which can be used to solve the exact version through boosting:

**Definition 2.2 (ANN).** *Let $c > 1$ and $r > 0$ be positive constants. In the $(c, r)$–**Approximate Nearest-Neighbors Problem (ANN)** we are given as input a set $S \subset M$ with $|S| = n$ and a sequence of queries $\{q_i\}_{i=1}^Q$ with $q_i \in M$. For each query $q_i$, if there exists $p \in B_S(q_i, r)$, we are required to output some point $p' \in B_S(q_i, cr)$. If $B_S(q_i, cr) = \emptyset$, we are required to output $\perp$. In the case where $B_S(q_i, r) = \emptyset \neq B_S(q_i, cr)$ we can either output a point from $B_S(q_i, cr)$ or $\perp$. Our algorithm should successfully satisfy these requirements with probability at least $2/3$.*

A prevalent method for solving ANN is Locality Sensitive Hashing (LSH). Intuitively, we seek a hash function that hashes close points together and far points apart with high probability.

**Definition 2.3 (Locality Sensitive Hashing, [35]).** *A hash family $\mathcal{H}$ of functions mapping $M$ to a set of buckets is called a $(c, r, p_1, p_2)$-**Locality Sensitive Hash Family (LSH)** if the following two conditions are satisfied:*

• If $x, y \in M$ have $||x - y|| \le r$, then $\Pr_{h \in \mathcal{H}}[h(x) = h(y)] \ge p_1$.

• If $x, y \in M$ have $||x - y|| \ge cr$, then $\Pr_{h \in \mathcal{H}}[h(x) = h(y)] \le p_2$.

where $p_1 \gg p_2$ are parameters in $(0, 1)$. We often assume that computing $h$ in a $d$–dimensional metric space requires $O(d)$ time.

Given a construction of a $(c, r, p_1, p_2)$–LSH for a metric space, we can solve the $(c, r)$–ANN problem by amplifying the LSH guarantees. This is done via an *"OR of ANDs"* construction: we sample $L = n^\rho$ hash functions $h_1, ..., h_L$ for $\rho \in (0, 1)$, each hashing $p \in \mathcal{M}$ to $\{0, 1\}^k$ by concatenating the outputs of $k = \lceil \log_{1/p_2} n \rceil$ "prototypical" LSH functions in $\mathcal{H}$ whose range is $\{0, 1\}$, as shown in [35]. This results in the following Theorem:

**Theorem 2.4.** *If a $d$–dimensional metric space admits a $(c, r, p_1, p_2)$–LSH family, then we can solve the $(c, r)$–ANN problem on it using $O(dn^{1+\rho})$ space and $O(dn^\rho)$ time per query, where*

$$\rho = \frac{\log p_1}{\log p_2}$$

As a byproduct of this construction, it is a standard observation that on any query and with high probability, no single bucket contains many outlier points outside of $B_S(q, cr)$ [35, 36].

**Lemma 2.5.** *Let $\mathcal{D} = (h_1, ..., h_L)$ be an LSH data structure consisting of $L = O(\log n \cdot n^\rho)$ hash functions that map the metric space $\mathcal{M}$ to $\{0, 1\}^k$, where $k = \lceil \log_{1/p_2} n \rceil$. Consider some query $q \in \mathcal{M}$ and let $S_{h_i(q)} := \{p \in S \mid h_i(q) = h_i(p)\}$ be the set of points in $S$ which are hashed in the same bucket as $q$ under $h_i$, for $i \in [L]$. Then, with high probability we have that:*

$$|S_{h_i(q)}| \le 3 \cdot |B_S(q, cr)|$$

*Furthermore, if $p \in B_S(q, r)$, then it is contained in at least one bucket with high probability. In other words, it is true that for some $i \in [L]$, with high probability, $p \in S_{h_i(q)}$.*

# 3 A Robust ANN Meta-Algorithm

In this section we give an adversarially robust ANN "meta-algorithm" that outperforms "for-all" algorithms when $d \gg Q$. For necessary theorems and notation on Differential Privacy, please refer to Appendix B.

## 3.1 Step 1: A Robust ANN "Decider"

Consider the following *decision ANN* problem:

> Given a point dataset $S \subset M$ and some radius parameter $r > 0$, on query $q \in M$ we wish to output 1 if and only if $B_S(q, r) \ne \emptyset$. Suppose $\mathcal{A}$ is an algorithm that can solve this problem with probability at least $9/10$ over an obliviously chosen input query sequence. Let us suppose that $\mathcal{A}$ first pre-processes $S$ to generate a data-structure $\mathcal{D}$, which it then uses to answer the queries. This algorithm is an *oblivious ANN decider*.

**Remark (Deciders are not Necessarily Robust!).** *It might be intuitively enticing to think that an oblivious decider is also robust. After all, an adversary providing queries knows the answer that they will, with high probability, receive from the algorithm, which is not true for the search version of the problem. However, the decisions alone of the algorithm can be more than enough to infer valuable information about its internal randomness. In particular, if the algorithm maintained a collection of hash tables in typical LSH-fashion, the attack of [43] can be performed on the decider equally well, especially if they have control over the input set $S$.*

We can design a $Q$–adversarially robust decider $\mathcal{A}_{\text{dec}}$ by using $\mathcal{A}$, while only increasing the space by a factor of $\sqrt{Q}$. Adhering to the framework of [38], we maintain $L = \widetilde{O}(\sqrt{Q})$ copies of the data

structures $\mathcal{D}_1, ..., \mathcal{D}_L$ generated by $\mathcal{A}$ using $L$ independent random strings, and then for each query $q$ we combine the answers of $\mathcal{A}$ privately.

As opposed to the original framework of [38], we do not need to use a private median algorithm, which simplifies the analysis and removes its dependency on that primitive. To keep the query time small, we utilize privacy amplification by subsampling (Theorem B.7).

---

**Algorithm 1** The robust decider $\mathcal{A}_{\text{dec}}$ (based on an oblivious decider $\mathcal{A}$)

1: **Inputs:** Random string $R = r_1 \circ r_2 \circ \cdots r_L$.
2: **Parameters**: Number of queries $Q$, number of copies $L$, number of sampled indices $k$.
3: Receive input dataset $S \subseteq U$ from the adversary, where $n = |S|$.
4: Initialize $\mathcal{D}_1, ..., \mathcal{D}_L$ where $\mathcal{D}_i \leftarrow \mathcal{A}(S)$ on random string $r_i$.
5: **for** $i = 1$ to $Q$ **do**
6:     Receive query $q_i$ from the adversary.
7:     $J_i \leftarrow$ Sample $k$ indices in $[L]$ with replacement.
8:     Let $a_{ij} \leftarrow \mathcal{D}_i(q_j) \in \{0, 1\}$ and $N_i := \frac{1}{k} |\{j \in J_i \mid a_{ij} = 1\}|$.
9:     Let $\widehat{N}_i = N_i + \text{Lap}\left(\frac{1}{k}\right)$.
10:     Output $\mathbb{1}[\widehat{N}_i > \frac{1}{2}]$

---

**Theorem 3.1.** *Let $\mathcal{A}$ be an oblivious decider algorithm for ANN that uses $s(n)$ space and $t(n)$ time per query. Let $\delta \in (0, 0.995)$ and suppose we set $L = 2400 \log^{1.5}(1/\delta) \cdot \sqrt{2Q}$ and $k = \log(Q/\delta)$. Then, the algorithm $\mathcal{A}_{dec}$ is an adversarially robust decider that succeeds with probability at least $1 - \Theta(\delta)$ using $s(n) \cdot \widetilde{O}\left(\sqrt{Q}\right)$ bits of space and $\widetilde{O}\left(t(n)\right)$ time per query.*

To prove Theorem 3.1 we argue that for all $i \in [Q]$, at least $\frac{8}{10}$ of the $k$ answers $a_{ij}$ are correct, even in the presence of adversarially generated queries. To do this, we first need to show that the algorithm is differentially private with respect to the input random strings $R$. Our analysis is included in full in Appendix C.

## 3.2 Step 2: From Robust Deciding to Robust Searching

In this section, we convert $\mathcal{A}_{\text{dec}}$ to an adversarially robust search algorithm. Let us assume for simplicity that $n = |S|$ is a power of two. We create a binary tree $\mathcal{T}$ over the entire input dataset $S$. Each node in the tree corresponds to a segment $[r_i, \ell_i]$ in $S$ that has size a power of 2. We create independent instances of $\mathcal{A}_{\text{dec}}$ in each node, each instance being initialized only for the dataset points inside that node's corresponding segment. When processing a query $q_i$, we first forward it to the root node. If it answers with a 1, it means that $B_S(q_i, r) \neq \emptyset$, so at least one of the two children will also return a 1. We can thus perform a root-to-leaf traversal to find and output an element of $S$ that lies in $f(S, q_i)$. On the other hand, if $f(S, q_i) = \emptyset$, the query at the root should tell us right away.

**Theorem 3.2.** *Let $\mathcal{A}$ be an oblivious ANN decider that uses $s(n)$ bits of space and answers each query in $t(n)$ time. Suppose that the space complexity $s(n)$ can be written as $s(n) = O(n^s)$ for some $s > 1$. Then, there exists an adversarially robust algorithm $\mathcal{A}'$ search ANN problem that uses $\widetilde{O}(\sqrt{Q} \cdot n^s)$ bits of space and has a query time of $\widetilde{O}(t(n))$.*

*Proof.* Before we execute our algorithm, we boost the probability of success for $\mathcal{A}_{\text{dec}}$ to $1 - \frac{2}{3n \log n}$ by scaling $\delta$ by $\frac{1}{n \log n}$ in Theorem 3.1. As a result, on any given query, *all* the copies of $\mathcal{A}_{\text{dec}}$ in any node are correct with probability at least $\frac{2}{3}$, by a union bound.

**Correctness** Fix some query $q_i$ that the adversary makes and suppose that $B_S(q_i, r) \neq \emptyset$. Let $p \in B_S(q_i, r)$ and consider the leaf $v$ in the binary tree containing $p$ in its segment. Then, in the path from the root $r(\mathcal{T})$ to $v$, we claim that all nodes have to answer 1 with high probability, regardless of the adversary's adaptivity, in which case we will definitely find $p$.

---

**Algorithm 2** A $Q$–adversarially robust search algorithm

---

1: Receive input dataset $S \subseteq U$ from adversary $\mathcal{B}$, where $n = |S|$.
2: Let $(s_1, ..., s_n)$ be an arbitrary ordering of $S$.
3: Create a rooted binary tree $\mathcal{T}$ over $[n]$ with $\log_2 n$ levels. Let $r(\mathcal{T})$ be the root of $\mathcal{T}$.
4: **for** each node $v = [r, \ell] \in \mathcal{T}$ **do**
5:     Initialize an independent copy $\mathcal{A}_{\text{dec}}^v$ of $\mathcal{A}_{\text{dec}}$ with input dataset $\{s_r, ..., s_\ell\} \subseteq S$.

6: **for** $i = 1$ to $Q$ **do**
7:     Receive query $q_i$ from $\mathcal{B}$.
8:     **if** $\text{QUERY}\left(A_{\text{dec}}^{r(\mathcal{T})}, q\right) = 0$ **then**
9:         Output $\perp$
10:    $v \leftarrow r(\mathcal{T})$.
11:    **while** $v \neq$ leaf **do**
12:        $v_r \leftarrow$ right child of $v$.
13:        **if** $\text{QUERY}(A_{\text{dec}}^{v_r}, q) = 1$ **then**
14:            $v \leftarrow v_r$
15:        **else**
16:            $v \leftarrow v_\ell$
17:    Output $s \in S$ where $\{s\}$ is the element corresponding to leaf $v$ in $\mathcal{T}$.

---

Let $p \in S_{[r,\ell]} := \{s_r, ..., s_\ell\}$ for some node $w = [r, \ell]$ on this path. Then, suppose, without loss of generality, that $p \in S_{[r, \frac{r+\ell}{2}]}$. Then we must have

$$B_{S_{[r, \frac{r+\ell}{2}]}}(q_i, r) \subseteq B_{S_{[r,\ell]}}(q_i, r)$$

196 That means that $B_{S_{[r, \frac{r+\ell}{2}]}}(q_i, r) \neq \emptyset$, which implies that this is also the case along the path from
197 $r(\mathcal{T})$ to $v$, by induction on the depth of the tree.

198 Since the error of each copy of $\mathcal{A}_{\text{dec}}$ is $\frac{2}{3n \log n}$, all copies in the tree are correct with probability at
199 least $\frac{2}{3}$, even against an adversarially generated query sequence. Hence, the aforementioned path
200 from the root to $v$ only has answers consisting of ones, meaning that we produce a point in $f(S, q_i)$.
201 On the other hand, if $B_S(q_i, r) = \emptyset$, the query to the root of the tree returns 0 with probability close
202 to 1, and so we correctly output $\perp$.

203 **Runtime** For the preprocessing, suppose a single copy of $\mathcal{A}_{\text{dec}}$ ran on a dataset $S$ of size $n$ takes
204 $O(\sqrt{Q} \cdot s(n) \cdot \text{polylog}(nQ)) = \widetilde{O}(\sqrt{Q} \cdot n^s)$ bits of space. Then the space of the search algorithm
205 can be bounded as:

$$\widetilde{O}\left(\sqrt{Q} \cdot \left[n^s + 2\left(\frac{n}{2}\right)^s + 4\left(\frac{n}{4}\right)^s + \cdots\right]\right)$$
$$= \widetilde{O}\left(\sqrt{Q} \cdot \sum_{i=0}^{\infty} 2^i \left(\frac{n}{2^i}\right)^s\right) = \widetilde{O}\left(n^s \sqrt{Q} \cdot \sum_{i=0}^{\infty} 2^{i-i-is}\right)$$
$$= \widetilde{O}\left(n^s \sqrt{Q} \cdot \sum_{i=0}^{\infty} (2^{-s})^i\right)$$
$$= \widetilde{O}\left(n^s \sqrt{Q}\right)$$

206 For the query time, we visit $\log_2 n$ vertices per query, so if we take $\widetilde{O}(t(n))$ time in total, which
207 completes the proof. $\qquad\square$

208 ### 3.3 Step 3: Building the Oblivious ANN Decider via LSH Sampling

209 In this section we build the oblivious ANN decider $\mathcal{A}$. For this we cannot simply run the familiar LSH
210 algorithm and output 1 whenever a point is found or a 0 otherwise. That is because LSH guarantees
211 to output a point from $B_S(q, cr)$ when $B_S(q, r) \neq \emptyset$ and is allowed to output a point from $B_S(q, cr)$

212 when $B_S(q, r) = \emptyset$. Our algorithm then would have no way of knowing which case a point in
213 $B_S(q, cr) \setminus B_S(q, r)$ is a signal for. It is clear that we need a different approach.

214 To create the decider algorithm, we modify the query algorithm that acts on top of the LSH data
215 structure as follows: Given $L = O(n^\rho \log n)$ hash functions within an LSH data structure, we sample
216 one uniformly at random. Let $h_i$ be the randomly sampled hash function, and let $h_i(q) \in \{0, 1\}^k$
217 be the bucket that query $q$ is hashed in. Let $S_{h_i(q)}$ be the set of points in $S$ that are also hashed to
218 $h_i(q)$. We sample one of those points uniformly at random. If we hit a point in $B_S(q, r)$ we output it.
219 Otherwise, we start the whole sampling process anew. If we haven't output 1 after $O(L \log n \cdot \frac{n(q,cr)}{n(q,r)})$
220 repetitions, then we output 0.

---

**Algorithm 3** Oblivious $(c, r)$–ANN via Sampling

1: **Input:** A dataset $S$ of $n$ points, parameters $c, r$.
2: Let $\rho = \rho(c)$ be the LSH parameter for this metric space.
3: Let $L \leftarrow O(n^\rho \log n)$ and $k = \Theta(\log n)$.
4: Initialize LSH data structures $\mathcal{D}_1, .., \mathcal{D}_L$ with hash functions $h_i : M \to \{0, 1\}^k$.
5: **for** each query $q \in M$ **do**
6:     **for** at most $O(L \log n \cdot \frac{n(q,cr)}{n(q,r)})$ iterations **do**
7:         Sample $i \in [L]$ uniformly at random. Sample a point $p$ from $S_{h_i(q)}$ uniformly at random.
8:         **if** $p \in B_S(q, r)$ **then**
9:             **Output** 1 for query $q$ and proceed to the next query.
10:     **Output** 0 and continue to the next query.

---

221 **Theorem 3.3.** *Assume that for each query $q \in \mathcal{M}$, we know that $\frac{n(q,cr)}{n(q,r)} \leq N$ when $n(q, r) \neq 0$.*
222 *Then, there exists an oblivious ANN decider algorithm using $\widetilde{O}(n^{1+\rho})$ bits of space and $\widetilde{O}(dn^\rho \cdot N)$*
223 *time per query.*

224 *Proof.* We calculate the probability that we successfully sample a point in $B_S(q, r)$. Let $\star$ be the
225 point we sample using one round of the above procedure, and let $i_\star$ be the hash function we pick.
226 Suppose for a point $p \in S$ that $B_p = \{i \in [L] \mid h_i(p) = h_i(q)\}$ is the set of hash functions that hash
227 $p$ the same as $q$. We have that:

$$\Pr[\star \in B_S(q, r)] = \sum_{p \in B_S(q,r)} \Pr[\star = p] = \sum_{p \in B_S(q,r)} \sum_{i=1}^{L} \Pr[\star = p \mid i_\star = i] \cdot \Pr[i_\star = i]$$

$$= \frac{1}{L} \sum_{p \in B_S(q,r)} \sum_{i=1}^{L} \Pr[\star = p \mid i_\star = i]$$

$$= \frac{1}{L} \sum_{p \in B_S(q,r)} \sum_{i \in B_p} \frac{1}{|S_{h_i(q)}|}$$

$$\geq \frac{1}{3L \cdot |B_S(q, cr)|} \sum_{p \in B_S(q,r)} |B_p|$$

$$\geq \frac{|B_S(q, r)|}{3L \cdot |B_S(q, cr)|}$$

228 where the last two inequalities follow with high probability from Theorem 2.5, which also implies
229 that $|B_p| \geq 1$ for all $p \in B_S(q, r)$. Now, the probability we do not sample a point $p \in B_S(q, r)$ after
230 $T$ trials is at most:

$$\left(1 - \frac{|B_S(q, r)|}{3L \cdot |B_S(q, cr)|}\right)^T \leq e^{-\frac{T \cdot |B_S(q,r)|}{3L \cdot |B_S(q,cr)|}} \leq \frac{1}{n}$$

231 because we set:

$$T = \frac{3L \cdot \log n \cdot |B_S(q, cr)|}{|B_S(q, qr)|}$$

Thus, we conclude that a point in $B_S(q,r)$ is returned with high probability. The space complexity of the algorithm is $O(Ln) = \widetilde{O}(n^{1+\rho})$, while the runtime per query is $O(\frac{L \log n \cdot n(q,cr)}{n(q,r)}) = \widetilde{O}(n^{\rho} \cdot \frac{n(q,cr)}{n(q,r)})$, as desired. $\qquad\square$

## 3.4   Step 4: Putting it all together

Combining the three previous steps, we reach the following result:

**Theorem 3.4.** *There exists an adversarially robust ANN algorithm using $\widetilde{O}(\sqrt{Q} \cdot n^{1+\rho})$ bits of space and $\widetilde{O}(dn^{\rho} \cdot \frac{n(q,cr)}{n(q,r)})$ time per query.*

# 4   Improvements in Robust and Fair ANN via Concentric LSH

The disadvantage of the sampling approach of Algorithm 3 – let us call it $\mathcal{L}_{\text{sampling}}$– is that its query runtime depends on the fraction $\frac{n(q,cr)}{n(q,r)}$, which could be really large. In this section we present a method for reducing the runtime under some mild assumptions.

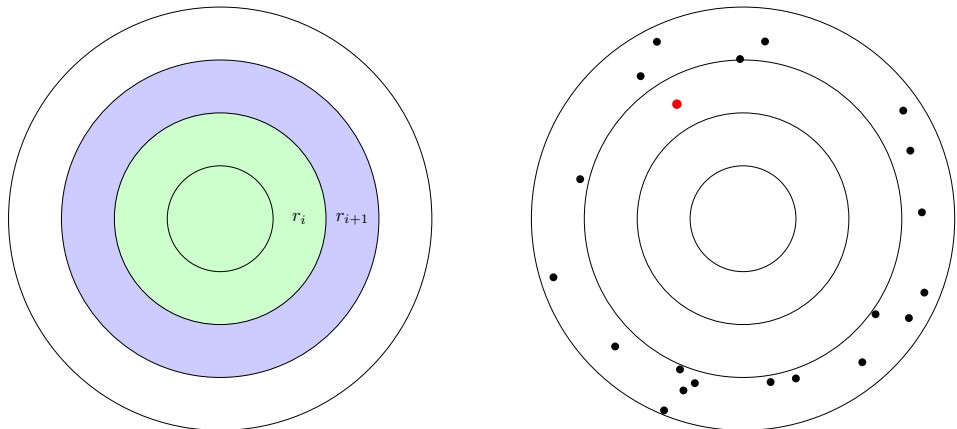

Figure 1: *Left*: In green lies the set $B_S(q, r_i)$, and blue represents the annulus that extends to $B_S(q, r_{i+1})$. *Right*: The degenerate case we explicitly disallow in Theorem 4.1.

Let $K$ be a parameter that we will soon resolve. Consider the following sequence of radii between $r$ and $cr$, interspersed so that the ratio between two consecutive ones is constant: $r_0 = r, r_1, ..., r_{K-1}, r_K = cr$ are defined as $r_i = c' \cdot r_{i-1}$ for $i \in [K]$, where $c' = c^{1/K}$. We create $K$ instances of $\mathcal{L}_{\text{sampling}}$ where the $i$-th instance $\mathcal{L}_i$, for $i \in [K]$, is initialized with parameters $(r_i, c')$. Our algorithm then runs each instance $\mathcal{L}_i$ to decide whether $B_S(q, r_i) \neq \emptyset$. If we observe an instance running for longer than $\Theta(n^{\rho + \frac{1}{K}})$ timesteps, we stop the execution and switch to the next instance.

As a result, our algorithm can decide whether $B_S(q, r_{K-1}) = \emptyset$ efficiently. However, there exists a degenerate case in which the time complexity is large, as shown in Figure 1. If $B_S(q, r_{K-2}) = \emptyset$, then it could be the case that $B_S(q, r_{K-1}) \neq \emptyset$ and $B_S(q, cr) \setminus B_S(q, r_{K-1})$ contains many points. This case is significantly rarer than the generic upper bound on $\frac{n(q,cr)}{n(q,r)}$ we assumed earlier, and we explicitly avoid it in our analysis:

**Assumption 1 ($K$-Degenerate Datasets).** *Suppose input dataset $S \in \mathcal{M}^n$ and parameter $K \in \mathbb{Z}_+$ are such that if $n(q, r_{K-2}) = 0$ and $n(q, r_{K-1}) \geq 1$ then $n(q, cr) \leq n^{1/K} \cdot n(q, r_{K-1})$.*

**Theorem 4.1.** *Assuming that dataset $S$ is not $K$-Degenerate for some integer $K \geq 2$, there exists an algorithm that on query $q \in \mathcal{M}$ outputs $\mathbb{1}[B_S(q,r) \neq \emptyset]$ with high probability, while using $O(n^{1 + \frac{1}{K} + \rho} \cdot \log n)$ bits of space overall and $O(dn^{\rho + 1/K} \log n)$ time per query.*

**Algorithm 4** Concentric Annuli LSH: An Improved Oblivious Decider

---

1: **Input:** A dataset $S$ of $n$ points, parameters $c, r, K$.

2: Let $c' \leftarrow c^{1/K}$ and $r_1 = r$.
3: **for** $i \in [K]$ **do**
4:     Initialize an independent copy $\mathcal{L}_i$ of $\mathcal{L}_{\text{sampling}}$ on $S$ with parameters $(r_i, c')$.
5:     Update $r_{i+1} = c' \cdot r_i$.
6: **for** each query $q \in M$ **do**
7:     **for** $i \in [K]$ **do**
8:         Let $(r_i, r_{i+1})$ be the sub-annulus $\mathcal{L}_i$ was initialized on.
9:         Run $\mathcal{L}_i$ for at most $100 \cdot n^{\rho+1/K}$ sample timesteps.
10:         **if** a point $p \in B_S(q, r_i)$ is found **then**
11:             **Output** 1 and continue to the next query.

12:     **Output** 0 and continue to the next query.

---

*Proof.* Our algorithm clearly runs in $\widetilde{O}(dn^{\rho+1/K})$ time because we truncate it at that timestep. The space complexity is implied by Theorem 3.3. Now, given a query $q$, first consider the case that for all $i \in \{0, 1, ..., K-1\}$ it is true that $n(q, r_i) \neq 0$. Then, suppose in that case that for all $i \in \{0, ..., K-1\}$ it holds that:

$$\frac{n(q, r_{i+1})}{n(q, r_i)} > n^{\frac{1}{K}}$$

Then, via a telescoping product we can write:

$$\frac{n(q, cr)}{n(q, r)} = \frac{n(q, r_1)}{n(q, r_0)} \cdot \frac{n(q, r_2)}{n(q, r_1)} \cdots \frac{n(q, r_{K-1})}{n(q, r_{K-2})} \cdot \frac{n(q, cr)}{n(q, r_{K-1})} > \left(n^{\frac{1}{K}}\right)^K = n$$

This is a contradiction because:

$$\frac{n(q, cr)}{n(q, r)} \leq n(q, cr) \leq n$$

Thus, Algorithm 4 will, in that case, terminate with output 1. Next, suppose that for all $0 \leq i \leq K-1$ we have that $n(q, r_i) = 0$. Then, we must have that $B_S(q, r_{K-1}) = \emptyset$, and Algorithm 4 correctly outputs 0 with high probability. Finally, let $j \leq K-1$ be the minimum index such that $n(q, r_i) > 0$. If $j < K-1$, then $\mathcal{L}_j$ will quickly and with high probability detect a point in $B_S(q, r_j) \subset B_S(q, r_{K-1})$ and output 1. If $j = K-1$, then $\mathcal{L}_{\text{sampling}}$ will find a point from $B_S(q, r_{K-1})$ in time $\widetilde{O}(n^\rho \cdot \frac{n(q, r_K)}{n(q, r_{k-1})}) = \widetilde{O}(n^{\rho+1/K})$ due to our structural assumption. $\qquad\square$

**Refining Theorem 3.4** Combining our concentric LSH approach with the meta-algorithm of the previous section, we arrive at our initially claimed result:

**Theorem 4.2.** *Let $K \geq 2$ be a parameter such that the input dataset $S$ is not $K$-degenerate. Then, there exists an adversarially robust ANN algorithm using $\widetilde{O}(\sqrt{Q} \cdot n^{1+\rho+1/K})$ bits of space and $\widetilde{O}(dn^{\rho+1/K})$ time per query.*

## 5 Conclusion

We studied efficient algorithms for Approximate Nearest Neighbor (ANN) queries against adaptive adversaries. Our approach uses a binary search tree to reduce the search task to a robust decision problem which we solve using techniques from Differential Privacy. We implement the decider with a sampling-based Locality-Sensitive Hashing (LSH) scheme accelerated by a concentric annuli construction. As a byproduct, this construction also yields a more efficient algorithm for exact fair ANN. Our approach constitutes a simple, universal framework for solving search problems efficiently against adaptive adversaries. Future work involves adapting our framework to other search problems and establishing computational lower bounds for robust search.

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

# A    Related Work

The challenge of designing algorithms robust to adversarial queries is well-studied, particularly in privacy and statistics [12, 49, 11], where Differential Privacy is a central tool for ensuring robustness [28, 27]. The question of adversarial robustness was formally introduced to streaming algorithms by Ben-Eliezer et al. [16], motivated by attacks on linear sketches [37], and has since inspired a long line of work on robustifying various streaming algorithms [38, 20, 44, 21, 50, 54, 15].

Our work is most directly inspired by the framework of Hassidim et al. [38], who used Differential Privacy to solve estimation problems robustly, and by Cherapanamjeri et al. [25], who applied this framework with low query time overhead. While we adapt a similar approach, their methods are fundamentally limited to estimation and don't extend to search problems like NNS, where the output must be a specific dataset element. The difficulty of robust search is further highlighted by Beimel et al. [14], who established lower bounds showing that robust algorithms for certain search problems are inherently slower than their oblivious counterparts, motivating our investigation.

Different works further reinforce the unique challenges of robust search. Work on robust graph coloring, for example, also requires techniques beyond simple noise addition due to its discrete output space [20, 13]. Our approach is also distinct from Las Vegas LSH constructions [46, 53]. While these methods guarantee no false negatives, they remain vulnerable to adversaries who can inflate their expected runtime [43]. Our focus, in contrast, is on robustifying traditional Monte Carlo algorithms.

Finally, our approach builds on the use of discretization and net-based arguments to achieve 'for-all' guarantees for ANN. This technique was previously used for robust distance estimation [23] and for ANN in conjunction with partition trees [24]. We contribute a simpler and more streamlined construction that offers a modest performance improvement over this prior work.

## A.1    Comparison with [32]

Our work was conducted concurrently and independently with that of Feng et al. [32], which also addresses adaptively solving search problems. The key similarities and differences are:

1. **Methodology:** The papers use fundamentally different approaches. Feng et al. employ a reduction to the private selection problem that is tightly coupled with the structure of DP noise. In contrast, we introduce a general "search-to-decision" meta-algorithm that treats the differentially private component as a black-box primitive.

2. **Assumptions and Performance:** The algorithms have different performance dependencies. Their runtime and space complexity both scale with a parameter $s$, which bounds the near-neighbor density: $|B_S(q, cr)| \leq s$. In contrast, our algorithm's space complexity has no such dependence on data density, making it strictly better in scenarios where $s$ is large. Our runtime is also independent of $s$, degrading only for **degenerate** datasets (Assumption 1)—a condition we argue is less restrictive. On the other hand, our algorithm increases the exponent by an additive factor of $\frac{1}{K}$, which in practice may be negligible[1]. While the scaling differs, both methods share a $\sqrt{Q}$ factor in space complexity from the use of DP.

Table 1: Comparison with Feng et al. [32]

| Metric | Our Algorithm | Feng et al. [32] |
|---|---|---|
| Query Time | $O(d \cdot n^{\rho + 1/K})^2$ | $O(d \cdot s \cdot n^\rho)$ |
| Space | $O(\sqrt{Q} \cdot n^{1+\rho+1/K})$ | $O(\sqrt{Q} \cdot s \cdot n^{1+\rho})$ |

# B    Review of Differential Privacy

Our work leans heavily on results from differential privacy, so we give the necessary definitions and results here.

---

[1]For instance, if $\rho = 0.25$ and $K = 100$.

[2]This holds for non-degenerate datasets as defined in Assumption 1.

## B.1 Definition of differential privacy

**Definition B.1 (Differential Privacy).** *Let $\mathcal{A}$ be any randomized algorithm that operates on databases whose elements come from some universe. For parameters $\varepsilon > 0$ and $\delta \in [0,1]$, the algorithm $\mathcal{A}$ is $(\varepsilon, \delta)$–differentially private (DP) if for any two neighboring databases $S \sim S'$ (ones that differ on one row only), the distributions on the algorithm's outputs when run on $S$ vs $S'$ are very close. That is, for any $S \sim S'$ and any subset of outcomes $T$ of the output space of $\mathcal{A}$ we have:*

$$\Pr[\mathcal{A}(S) \in T] \le e^{\varepsilon} \cdot \Pr[\mathcal{A}(S') \in T] + \delta$$

## B.2 The Laplace Mechanism and its properties

**Theorem B.2 (The Laplace Mechanism, [30]).** *Let $f : X^* \to \mathbb{R}$ be a function. Define its sensitivity $\ell$ to be an upper bound to how much $f$ can change on neighboring databases:*

$$\forall S \sim S' : \quad |f(S) - f(S')| \le \ell$$

*The algorithm that on input $S \in X^*$ returns $f(S) + Lap\left(\frac{\ell}{\varepsilon}\right)$ is $(\varepsilon, 0)$–DP, where*

$$Lap(\lambda; x) := \frac{1}{2\lambda} \exp\left(-\frac{|x|}{\lambda}\right)$$

*is the Laplace Distribution over $\mathbb{R}$.*

We will make use of the following concentration property of the Laplace Distribution:

**Lemma B.3.** *For $m \ge 1$, let $Z_1, ... Z_m \sim Lap(\lambda)$ be iid random variables. We have that:*

$$\Pr\left[\max_{i=1}^{m} Z_i > \lambda(\ln(m) + t)\right] \le e^{-t}$$

## B.3 Properties of differential privacy

Differential Privacy has numerous properties that are useful in the design of algorithms. The following theorem is known as "advanced adaptive composition" and describes a situation when DP algorithms are linked sequentially in an adaptive way.

**Theorem B.4 (Advanced Composition, [31]).** *Suppose algorithms $\mathcal{A}_1, ..., \mathcal{A}_k$ are $(\varepsilon, \delta)$–DP. Let $\mathcal{A}'$ be the adaptive composition of these algorithms: on input database $x$, algorithm $\mathcal{A}_i$ is provided with $x$, and, for $i \ge 2$, with the output $y_{i-1}$ of $\mathcal{A}_{i-1}$. Then, for any $\delta' \in (0,1)$, Algorithm $\mathcal{A}$ is $(\widetilde{\varepsilon}, \widetilde{\delta})$–DP with:*

$$\widetilde{\varepsilon} = \varepsilon \cdot \sqrt{2k \ln(1/\delta')} + 2k\varepsilon^2 \text{ and } \widetilde{\delta} = k\delta + \delta'$$

The next theorem dictates that post-processing the output of a DP algorithm cannot degrade its privacy guarantees, as long as the processing does not use information from the original database.

**Theorem B.5 (DP is closed under Post-Processing).** *Let $\mathcal{A} : U^n \to Y^m$ and $\mathcal{B} : Y^m \to Z^r$ be randomized algorithms, where $U, Y, Z$ are arbitrary sets. If $\mathcal{A}$ is $(\varepsilon, \delta)$–DP, then so is the composed algorithm $\mathcal{B}(\mathcal{A}(\cdot))$.*

The following theorem showcases the power of DP algorithms in learning.

**Theorem B.6 (DP and Generalization, [11, 29]).** *Let $\varepsilon \in (0, 1/3)$ and $\delta \in (0, \varepsilon/4)$. Let $\mathcal{A}$ be a $(\varepsilon, \delta)$–DP algorithm that operates on databases in $X^n$ and outputs $m$ predicate functions $h_i : X \to \{0, 1\}$ for $i \in [m]$. Then, if $D$ is any distribution over $X$ and $S$ consists of $n \ge \frac{1}{\varepsilon^2} \cdot \log\left(\frac{2\varepsilon m}{\delta}\right)$ iid samples from $D$, we have for all $i \in [m]$ that:*

$$\Pr_{\substack{S \sim D^n \\ h_i \leftarrow \mathcal{A}(S)}} \left[\left|\frac{1}{|S|} \sum_{x \in S} h_i(x) - \mathbb{E}_{x \sim D}[h_i(x)]\right| \ge 10\varepsilon\right] \le \frac{\delta}{\varepsilon}$$

In other words, a privately generated predicate is a good estimator of its expectation under any distribution on the input data. A final property of privacy that we will use is a boosting technique through sub-sampling:

**Theorem B.7 (Privacy Amplification by Subsampling, [19, 25]).** *Let $\mathcal{A}$ be an $(\varepsilon, \delta)$–DP algorithm operating on databases of size $m$. For $n \geq 2m$, consider an algorithm that for input a database of size $n$, it subsamples (with replacement) $m$ rows from the database and runs $\mathcal{A}$ on the result. Then this algorithm is $(\varepsilon', \delta')$–DP for*

$$\varepsilon' = \frac{6\varepsilon m}{n} \ \text{ and } \ \delta' = \exp\left(\frac{6\varepsilon m}{n}\right) \cdot \frac{4m}{n} \cdot \delta$$

# C   Proof of Theorem 3.1

In this section we include a formal proof of Theorem 3.1 on the construction of a robust decider algorithm.

**Lemma C.1.** *Let $\varepsilon = 0.01$ and $\delta \in (0, 0.995)$. Algorithm $\mathcal{A}_{dec}$ is $(\varepsilon, \delta)$–DP with respect to the string of randomness $R$.*

*Proof.* We analyze the privacy of the algorithm $\mathcal{A}_{\text{dec}}$ given in Algorithm 1 with respect to the string of randomness $R$, which we interpret as its input. Suppose we let

$$\varepsilon' = \frac{\varepsilon}{2\sqrt{2Q\ln(1/\delta)}}$$

For all $i \in [Q]$, we claim that the response to query $q_i$ is $(\varepsilon', 0)$–DP with respect to $R$. This is because the statistic $N_i$ defined in Line 8 of Algorithm 1 has sensitivity $1/k$ and therefore by Theorem B.2, after applying the Laplace mechanism in Line 9, we have that releasing $\widehat{N}_i$ is $(1, 0)$–DP with respect to the strings $R$. The binary output based on comparing $\widehat{N}_i$ with the constant threshold $1/2$ is still $(1, 0)$-DP by post-processing (Theorem B.5).

Since $L \geq 2k$, using the amplification by sub-sampling property (Theorem B.7), we get that each iteration is $(\varepsilon', 0)$–DP, because for large enough $Q$ we have:

$$\frac{6k}{L} = \frac{6\varepsilon \log \frac{1}{\delta} + 6\varepsilon \log Q}{24 \cdot \log \frac{1}{\delta}\sqrt{2Q\ln\left(\frac{1}{\delta}\right)}} < \frac{2\varepsilon}{4\sqrt{2Q\ln\left(\frac{1}{\delta}\right)}} = \varepsilon'$$

Finally, by adaptive composition (Theorem B.4), after $Q$ adaptive steps our resulting algorithm is $(\varepsilon'', \delta)$-DP where:

$$\varepsilon'' = \varepsilon'\sqrt{2Q\ln\left(\frac{1}{\delta}\right)} + Q(\varepsilon')^2 = \frac{\varepsilon}{2} + \frac{\varepsilon^2}{4\ln\left(\frac{1}{\delta}\right)} \leq \varepsilon$$

for $\varepsilon \leq 2\ln \delta^{-1}$, which is satisfied for $\delta \in (0, 0.995)$. Thus, Algorithm $\mathcal{A}_{\text{dec}}$ is $(\varepsilon, \delta)$–DP with respect to its inputs – the random strings $R$. $\qquad\square$

Next, we show that a majority of the data structures $\mathcal{D}_i$ output accurate verdicts with high probability, even against adversarially generated queries.

**Lemma C.2.** *With probability at least $1 - \delta$, for all $i \in [Q]$, at least $0.8L$ of the answers $a_{ij}$ are accurate responses to the decision problem with query $q_i$.*

*Proof.* The central idea of the proof, as it appeared in [38], is to imagine the adversary $\mathcal{B}$ as a post-processing mechanism that tries to guess which random strings lead $\mathcal{A}$ to making a mistake.

Imagine a wrapper *meta-algorithm $\mathcal{C}$*, outlined as Algorithm 5, that takes as input the random string $R = r_1 \circ r_2 \circ \cdots \circ r_L$, which is generated according to some unknown, arbitrary distribution $\mathcal{R}$. This

523 algorithm $\mathcal{C}$ simulates the game between $\mathcal{A}_{\text{dec}}$ and $\mathcal{B}$: It first runs $\mathcal{B}$ to provide some input dataset
524 $S \subseteq U$ to $\mathcal{A}_{\text{dec}}$, which is seeded with random strings in $R$. Then, $\mathcal{C}$ uses $\mathcal{B}$ to query $\mathcal{A}_{\text{dec}}$ adaptively
525 with queries $(q_1, ..., q_Q)$. At the same time, it simulates $\mathcal{A}_{\text{dec}}$ to receive answers $a_1, ..., a_Q$ that are
526 fed back to $\mathcal{B}$. By Theorem C.1, the output $(a_1, ..., a_Q)$ is produced privately with respect to $R$,
527 regardless of how the adversary makes their queries.

528 At every step $i$, once $\mathcal{B}$ has provided $\vec{q}_i = (q_1, ..., q_i)$ and has gotten back $i$ answers $(a_1, ..., a_i)$
529 from $\mathcal{A}_{\text{dec}}$, our meta-algorithm $\mathcal{C}$ *post-processes* this output history $\{(q_j, a_j)\}_{j=1}^i$ to generate a
530 predicate $h_{\vec{q}_i} : \{0,1\}^* \to \{0,1\}$. This predicate tells which strings $r \in \{0,1\}^*$ lead algorithm $\mathcal{A}$
531 to successfully answer query prefix $\vec{q}_i$ on input dataset $S$, in the decision-problem regime. More
532 formally[3]:

$$h_{\vec{q}_i}(r) := \bigwedge_{1 \leq j \leq i} \{\mathcal{A}(r)(S, q_j) = \mathbb{1}\left[B_S(q_j, \bar{r}) \neq \emptyset\right]\} \tag{2}$$

---

**Algorithm 5** The meta-algorithm $\mathcal{C}$, ran for $i$ steps

---

1: **Inputs:** Random string $R = r_1 \circ r_2 \circ \cdots r_L$, descriptions of Algorithms $\mathcal{A}_{\text{dec}}$ and $\mathcal{B}$.
2: Simulate $B$ to obtain a dataset $S \subset U$.
3: Initialize $\mathcal{A}_{\text{dec}}$ with random strings $(r_1, ..., r_L)$ and the dataset $S$.
4: **for** $i \in Q$ **do**
5: $\quad$ Simulate $\mathcal{B}$ to produce a query $q_j$ based on the prior history of queries and answers.
6: $\quad$ Simulate $\mathcal{A}$ on query $q_j$ to produce an answer.
7: $\quad$ Compute (via post-processing of query/answer history) predicate $h_{\vec{q}_i}(\cdot)$ from Equation 2.
8: **Output** $(h_{\vec{q}_1}, ..., h_{\vec{q}_Q})$.

---

533 Generating these predicates is possible because $h_{\vec{q}_i}$ only depends on $\vec{q}_i$, which is a substring of the
534 output history that $\mathcal{C}$ has access to. As a result, $\mathcal{C}$ can produce $h_{\vec{q}_i}$ by (say) calculating its value for
535 each value of $R$ exhaustively[4]. Because $\mathcal{C}$ is only allowed to post-process the query/answer vector
536 $(q_1, a_1, ..., q_i, a_i)$, the output predicate $h_{\vec{q}_i}$ is also generated in a $(\varepsilon, \delta)$–DP manner with respect to
537 $r_1, ..., r_L$, by Theorem B.5.

538 Given these $Q$ privately generated predicates, and since $L > \frac{1}{\varepsilon^2} \log \frac{2\varepsilon Q}{\delta}$ for large enough $Q$, by the
539 generalization property of DP (Theorem B.6) we have that with probability at least $1 - \frac{\delta}{\varepsilon} = 1 - \Theta(\delta)$
540 it holds for any distribution $\mathcal{R}$ and for all $i \in [Q]$ that:

$$\left| \mathbb{E}_{r \sim \mathcal{R}} [h_{\vec{q}_i}(r)] - \frac{1}{L} \sum_{j=1}^L h_{\vec{q}_i}(r_j) \right| \leq 10\varepsilon = \frac{1}{10} \tag{3}$$

541 But if $\mathcal{R}$ is the uniform distribution, then $\mathbb{E}_{r \sim \mathcal{R}} [h_{\vec{q}_i}(r)]$ is simply the probability that $\mathcal{A}_2$ gives an
542 accurate answer on the *fixed* query sequence $\vec{q}_i$. Since $\mathcal{A}$ is an oblivious decider, Equation 3 implies
543 that:

$$\mathbb{E}_{r \sim \mathcal{R}} [h_{\vec{q}_i}(r)] \geq \frac{9}{10} \tag{4}$$

544 Further, $\frac{1}{L} \sum_{j=1}^L h_{\vec{q}_i}(r_j)$ is the fraction of random strings that lead $\mathcal{A}_2$ to be correct. Thus, by
545 Equation 4, this fraction is at least $\left(\frac{9}{10} - \frac{1}{10}\right) L = 0.8L$ for all $i \in [Q]$. $\qquad \square$

546 We are now ready to prove the main theorem of this section.

547 *Proof of Theorem 3.1.* Let us condition on the event that Theorem C.2 holds, which happens with
548 probability at least $1 - \Theta(\delta)$. Then, for all $i \in [Q]$, $N_i$ is either at least 0.8, when $B_S(q_j, \bar{r}) \neq \emptyset$, or

---

[3]We replace the radius parameter $r$ with $\bar{r}$ briefly in this argument. The symbol $r$ is reserved for an arbitrary random string.

[4]We assume $\mathcal{C}$ has unbounded computational power.

at most $1 - 0.8 = 0.2$, otherwise. By Theorem B.3, we require that the maximum Laplacian noise not exceed 0.2 with high probability:

$$\Pr\left[|Z_i| > 0.2\right] = \Pr\left[|Z_i| > \frac{1}{k}\left(\ln(1) + 0.2k\right)\right] \leq e^{-0.2k} \tag{5}$$

Since our threshold for deciding is $\widehat{N}_i := N_i + Z_i \geq 0.5$, we can see that setting $k = \Omega(\log(Q/\delta))$ will make the probability in Equation 5 at most $\frac{\delta}{Q}$, implying, by union bound, that $\mathcal{A}_{\text{dec}}$ outputs the correct answer at every timestep $i \in [Q]$ with high probability. $\qquad\square$

## D    Improved Robust ANNS Algorithms with $\forall$ guarantees

In this section, we will discuss another path to adversarial robustness for search problems –providing a *for-all* guarantee. We will focus on the ANN problem for this section, due to its ubiquity and importance, as well as its amenity to the techniques we discuss.

### D.1    A *For-all* guarantee in the Hamming cube

We present the Hamming Distance ANN case first because it is the most natural *for-all* guarantee one can give. This is because the space we are operating over is discrete, and we can easily union-bound over all possible queries and only incur a cost polynomial to the dimension $d$ of the metric space.

**Theorem D.1.** *There exists an adversarially robust algorithm solving the $(c, r)$–ANN problem in the $d$–dimensional Hamming Hypercube that can answer every possible query correctly with probability at least $1 - 1/n^2$. The space requirements are $\widetilde{O}(d \cdot n^{1+\rho+o(1)})$, and the time required per query is $\widetilde{O}(d^2 \cdot n^\rho)$, where $\rho = 1/c$.*

*Proof.* First, let us recall the standard LSH in the Hamming Hypercube: We are given a point set $S \subseteq \{0, 1\}^d$ with $|S| = n$. We receive queries $q \in \{0, 1\}^d$. Our Locality Sensitive Hash family $\mathcal{H}$ is defined as follows: Pick some coordinate $i \in [d]$ and hash $x \in \{0, 1\}^d$ according to $x_i \in \{0, 1\}$. This function $h$ acts as a hyperplane separating the points in the hypercube into two equal halves, depending on the $i$-th coordinate. Sampling $h$ uniformly at random from $\mathcal{H}$ is equivalent to sampling $i \in [d]$ uniformly at random. We can easily see that $\mathcal{H}$ is an $(r, cr, p_1, p_2)$–LSH family, as:

$$\Pr_{h \sim \mathcal{H}}\left[h(p) = h(q)\right] = \frac{d - ||p - q||}{d} = \begin{cases} \geq 1 - \frac{r}{d} := p_1, & \text{when } ||p - q|| \leq r \\ \leq 1 - \frac{cr}{d} := p_2, & \text{when } ||p - q|| \geq cr \end{cases}$$

We now go through the typical amplification process for LSH families [33]. Instead of sampling just one coordinate, we sample $k$. And instead of sampling just one hash function, we sample $L$ different ones $h_1, ..., h_L \in \mathcal{H}^k$ and require that a close point collides with $q$ at least once. With this scheme, we know that if we fix $q \in \{0, 1\}^d$ and $p \in B_S(q, r)$ we have:

$$\Pr\left[\exists i \in [L] \,:\, h_i(p) = h_i(q)\right] \geq 1 - (1 - p_1^k)^L$$

Furthermore, if $||p - q|| \geq cr$, we must have:

$$\Pr\left[\exists i \in [L] \,:\, h_i(q) = h_i(p)\right] \leq L p_2^k$$

Now, we want to guarantee that with high probability there doesn't exist any query $q \in \{0, 1\}^d$ such that for all points $p \in B_S(q, r)$ we have $h_i(q) \neq h_i(p)$ for all $i \in [L]$. In other words, we want:

$$\Pr\left[\exists q \in \{0, 1\}^d \,:\, \forall p \in B_S(q, r)\, \forall i \in [L] : h_i(p) \neq h_i(q)\right] \leq \frac{1}{n}$$

We can use the union bound to get:

$$\Pr\left[\exists q \in \{0, 1\}^d \,:\, \forall p \in B_S(q, r)\, \forall i \in [L] : h_i(p) \neq h_i(q)\right]$$
$$\leq \sum_{q \in \{0, 1\}^d} \Pr\left[\forall p \in B_S(q, r)\, \forall i \in [L] : h_i(p) \neq h_i(q)\right]$$

So it suffices to establish that for fixed $q \in \{0, 1\}^d$ we have:

$$\Pr\left[\forall p \in B_S(q, r) \, \forall i \in [L] : h_i(p) \neq h_i(q)\right] \leq \frac{1}{n2^d}$$

We can weaken this statement and union-bound as follows:

$$
\begin{aligned}
\Pr\left[\forall p \in B_S(q, r) \, \forall i \in [L] : h_i(p) \neq h_i(q)\right] &\leq \Pr\left[\exists p \in B_S(q, r) \; \not\exists i \in [L] : h_i(p) = h_i(q)\right] \\
&\leq \sum_{p \in B_S(q,r)} \Pr\left[\not\exists i \in [L] : h_i(p) = h_i(q)\right] \\
&\leq |B_S(q, r)| \cdot (1 - p_1^k)^L \\
&\leq n(1 - p_1^k)^L
\end{aligned}
$$

So it suffices to require that:

$$(1 - p_1^k)^L \leq \frac{1}{n^2 2^d} \tag{6}$$

On the other hand, the expected number of points in $S \setminus B_S(q, cr)$ that we will see in the same buckets as $q$ is:

$$\mathbb{E}\left[|p \in S \setminus B_S(q, cr) \mid \exists i \in [L] \,:\, h_i(p) = h_i(q)|\right] = \sum_{p \in S \setminus B_S(q,cr)} \Pr\left[\exists i \in [L] \mid h_i(p) = h_i(q)\right] \tag{7}$$

$$\leq nL p_2^k \tag{8}$$

We can now combine Equation 6 and Equation 8 to work out the values of $k$ and $L$. First, we want to get $O(L)$ time in expectation, so we require $p_2^k \leq 1/n$, which gives:

$$k \geq \log_{1/p_2}(n)$$

Now, let $p_1 = p_2^\rho$. Substituting, we resolve the value of $L$ as:

$$L \geq n^\rho d \log n$$

With that in place, we can see that our algorithm takes $O(L)$ time with high probability. Indeed, let $X$ be the number of points in $S \setminus B_S(q, cr)$ that are hashed to some common bucket with $q$. Using a simplified Chernoff bound, we have that:

$$\Pr\left[X \geq 10L\right] \leq 2^{-10L} = \frac{1}{n^{10dn^\rho}} \ll \frac{1}{n^{\Omega(1)}}$$

which implies that our runtime per query is $O(L)$ with high probability. As for the value of the constant $\rho$ we have by definition that:

$$\rho := \frac{\log p_1}{\log p_2} = \frac{\log\left(1 - \frac{r}{d}\right)}{\log\left(1 - \frac{cr}{d}\right)} \approx \frac{1}{c}$$

Overall, evaluating our hash function requires $\widetilde{O}(\log n)$ time, and evaluating distances between points requires $O(d)$ time. We maintain $O(d \cdot n^\rho \log n)$ hash tables, meaning that on a single query we spend $O(d^2 \cdot n^\rho \log n)$ time. For pre-processing, apart from storing the entire dataset in $dn$ space, we take $O(d \cdot n^{1+\rho+o(1)})$ space to construct our data structure. $\square$

### D.1.1 Improving the query runtime via sampling

We can improve the dependency on $d$ for the query runtime by using sampling to find a good bucket. The following theorem encapsulates this finding, reducing the runtime complexity by a factor of $d$:

**Theorem D.2.** *There exists an adversarially robust algorithm solving the $(c, r)$–ANN problem in the $d$–dimensional Hamming Hypercube that can answer all possible queries correctly with probability at least $1 - 1/n^2$. The space requirements are $\widetilde{O}(d \cdot n^{1+\rho+o(1)})$ and the time required per query is $\widetilde{O}(d \cdot n^\rho)$, where $\rho = 1/c$.*

*Proof.* From our analysis above, we know that we take $L = n^\rho \cdot d \log n$ different hash functions. Consider some query $q$. We analyze the expected number of buckets that contain some point $p \in B_S(q, r)$. Let $X_q$ be a random variable representing the number of buckets $i \in [L]$ for which some point in $B_S(q, r)$ lies in bucket $i$. Define the following indicator random variable:

$$\mathbb{1}_i = \begin{cases} 1, & \text{if some point } p \in B_S(q, r) \text{ lies in bucket } i \in [L] \\ 0, & \text{otherwise} \end{cases}$$

By linearity of expectation, we can now write:

$$\begin{aligned}
\mathbb{E}[X_q] &= \sum_{i=1}^{L} \Pr[\mathbb{1}_i = 1] \\
&= \sum_{i=1}^{L} \Pr\left[ \bigcup_{p \in B_S(q, r)} \{h_i(p) = h_i(q)\} \right] \\
&\geq L \cdot p_1^k \\
&= L \cdot (p_2)^{\rho k} \\
&\geq \frac{L}{n^\rho} \\
&= d \log n
\end{aligned}$$

By using the Chernoff bound, we can see that with high probability, $X_q$ is close to its expectation:

$$\Pr\left[ X_q \leq \frac{1}{2} d \log n \right] \leq e^{-\frac{d \log n}{8}} = \frac{1}{n^{d/8}} \ll \frac{1}{n}$$

Let us, then, condition on $X_q > \frac{1}{2} d \log n$. On query time, we can simply sample $m = \Theta(n^\rho \log n)$ buckets uniformly at random from $[L]$. We know that with probability at least $\frac{d \log n}{2 n^\rho d \log n} = \frac{1}{2 n^\rho}$, a single randomly selected bucket contains some point from $B_S(q, r)$. So, for all $m$ of the selections to not contain such a point, the probability is at most:

$$\left( 1 - \frac{1}{n^\rho} \right)^{n^\rho \log n} \leq e^{-\log n} = \frac{1}{n}$$

So, with probability at least $1 - \frac{1}{n}$ we find a bucket containing a good point. Since, with high probability, the number of points in $P \setminus B_S(q, cr)$ in any bucket are $O(L)$, we see that this sampling method improves the query runtime to $O(n^\rho \log n)$. $\qquad\square$

### D.1.2 Utilizing the optimal LSH algorithm

Our earlier exposition used the original LSH construction for the Hamming Hypercube [40] that achieves $\rho = 1/c$. We can also use the state-of-the-art approach from [6] that achieves $\rho = \frac{1}{2c-1}$ in place of Theorem D.1. This slightly improves the exponent on $n$:

**Theorem D.3.** *There exists an adversarially robust algorithm solving the $(c, r)$–ANN problem in the $d$–dimensional Hamming Hypercube that can answer all possible queries correctly with probability at least $0.99$. The space complexity is $O(d \cdot n^{1+\rho+o(1)})$, and the time required per query is $O(d \cdot n^\rho)$, where $\rho = \frac{1}{2c-1}$. These runtime guarantees hold with high probability.*

The analysis is identical, so we will not repeat it again: Since the algorithm succeeds with constant probability, and we want it to succeed on all $2^d$ possible queries, we boost its success probability to $1 - \frac{1}{100 \cdot 2^d}$. This way, after the union bound, any query succeeds with probability at least $0.99$. Furthermore, the analysis of the sampling algorithm for improving the query runtime in Theorem D.2 also remains the same. All that changes between using the standard Hamming norm LSH as opposed to the optimal one is the ratio $\rho := \frac{\log p_1}{\log p_2}$.

## D.2 Discretization of continuous spaces through metric coverings

The *for-all* algorithm we presented as Theorem D.2 cannot be applied outside of discrete spaces, however, because the key to our analysis was the union bound over all the possible queries.

To simulate a similar argument for solving ANN in continuous, $\ell_p$ spaces, we can consider a strategy of discretizing the space. We place special "marker" points and guarantee that some version of the ANN problem is solvable around them. Then, when a query comes in, we find its corresponding marker point, and solve the ANN problem for it. We show that the answer we get is valid for the original query as well, so long as the "neighborhood" around the marker points is small enough. A similar strategy and covering construction appeared in [24], although they did not make algorithmic use of the ability to project any query point to the covering set. Instead, their algorithm deems it sufficient to be successful on every point on just the covering set.

### D.2.1 Metric coverings in continuous spaces

To initiate our investigation, we need the definition of a *metric covering*:

**Definition D.4.** *Consider a metric space $\mathcal{M} = (\mathbb{R}^d, ||\cdot||_p)$ with metric $\mu$. Let $U \subset \mathbb{R}^d$ be a bounded subset. A set $\widehat{S} \subseteq \mathbb{R}^d$ is called an $\Delta$-**covering** of $U$ if for all $q \in U$ there exists some $\widehat{s} \in \widehat{S}$ such that*

$$||q - \widehat{s}||_p \leq \Delta$$

Suppose that $U$ is a bounded subset of $\mathbb{R}^d$. We can construct the following the following $\Delta$-covering of $U$: Let $C := \sup_{x \in U} ||x||_\infty$ and suppose $\{u_i\}_{i=1}^d$ is an orthonormal basis spanning $U$. We know that $||x||_\infty \leq C$ for all $x \in U$, so let us define:

$$\widehat{S} = \sum_{i=1}^d \widehat{\alpha}_i u_i, \quad \text{where}$$

$$\widehat{\alpha}_i \in \{-C, -C + \varepsilon, ..., C - \varepsilon, C\}$$

for some choice of $\varepsilon$ that we will decide later. This is a standard construction for $\ell_2$ that we now extend to $\ell_p$ [48]. As defined, we have:

$$\left|\widehat{S}\right| = \left(\frac{2C}{\varepsilon}\right)^d$$

Now, fix some $q \in U$. We can write:

$$q = \sum_{i=1}^d \alpha_i u_i$$

For all $i \in [d]$, let $\widehat{\alpha}_i$ be such that $\alpha_i \in \widehat{\alpha}_i \pm \varepsilon$. Let $\widehat{s} := \sum_{i=1}^d \widehat{\alpha}_i u_i$. Now we have that:

$$||q - \widehat{s}||_p^p = \left|\left|\sum_{i=1}^d (\alpha_i - \widehat{\alpha}_i)u_i\right|\right|_p^p = \sum_{i=1}^d |\alpha_i - \widehat{\alpha}_i|^p \leq d\varepsilon^p$$

Now, let us set:

$$\varepsilon = \frac{\Delta}{d^{1/p}} \implies ||q - \widehat{s}||_p \leq \Delta$$

Our construction thus has size:

$$|\widehat{S}| = \left(\frac{2Cd^{1/p}}{\Delta}\right)^d$$

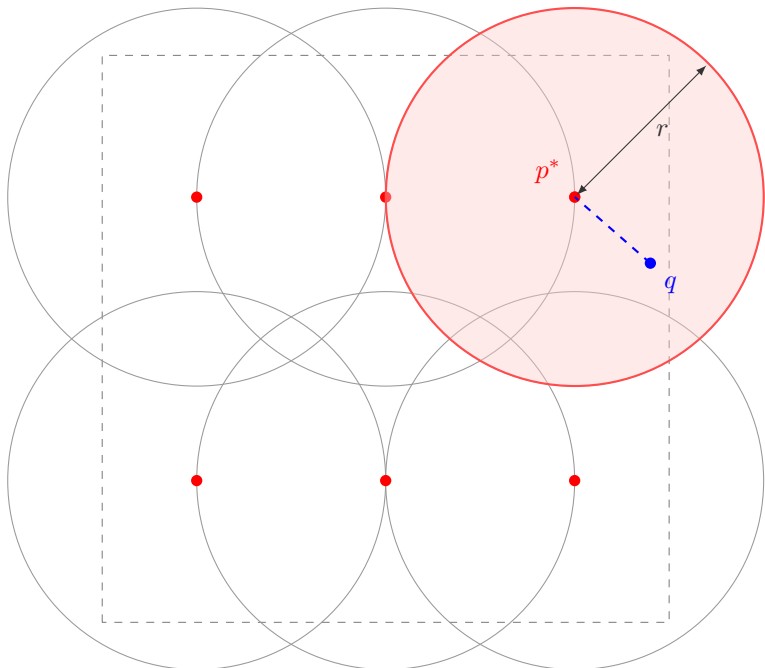

Figure 2: An illustration of an $r$-covering.

 ## D.2.2 The robust ANN algorithm

657 With this construction in mind, our algorithm for robust $(c, r)$–ANN in $\ell_p$ space follows as Algorithm
658 6. The algorithm remains agnostic to the specific LSH data structure that could be used to solve ANN
659 in $\ell_p$ metric spaces obliviously [22, 26], but assumes that the success probability over a set of queries
660 in that data structure can be boosted by increasing the number of hash functions taken. This was the
661 case for the Hamming norm as well.

---

**Algorithm 6** Robust $\ell_p$ ANN through discretization

---

1: *Parameters:* Max-norm $C$, runtime/accuracy tradeoff $\Delta > 0$, LSH parameters $c, r > 0$.
2: Receive point dataset $S \subset U$ with $|S| = n$ from the adversary.
3: Let $\widehat{S}$ be a $\Delta$-covering of $U$ as constructed in Section D.2.1, and let $c' \leftarrow \frac{cr - \Delta}{r + \Delta}$.
4: Initialize an LSH data structure $\mathcal{D}$ for solving $(c', r + \Delta)$–ANN that answers all queries in $\widehat{S}$
   correctly with high probability.
5: **while** Adversary provides queries **do**
6:     Receive query $q \in U$ from the adversary.
7:     Find $\widehat{s} \in \widehat{S}$ such that $||q - \widehat{s}||_p \leq \Delta$.
8:     Query $\mathcal{D}$ on $\widehat{s}$ and output whatever it outputs.

---

662

663 **Theorem D.5.** *There exists an adversarially robust algorithm solving the $(c, r)$–ANN problem in the*
664 $(\mathbb{R}^d, \ell_p)$ *metric space that can answer an unbounded number of adversarial queries. Assuming that*
665 *the input dataset and the queries are all elements of $U = \{x \in \mathbb{R}^d \mid ||x||_p \leq C\}$ for some $C > 0$,*
666 *the pre-processing space is $\widetilde{O}(nT)$ and the time per query is $\widetilde{O}(T)$, where:*

$$T = O\left[ d \cdot n^{\rho'} \log\left( \frac{Cd^{1/p}}{cr} \right) \right] \tag{9}$$

667 *where:*

$$\rho' = \frac{(10 + c)^2}{161c^2 - 20c - 100}$$

*Proof.* First, to argue for correctness, let $q$ be any query. Suppose there exists some point $x \in S$ with $||x - q||_p \leq r$. Then, by triangle inequality it holds that:

$$||x - \widehat{s}||_p \leq ||x - q||_p + ||\widehat{s} - q||_p \leq \Delta + r$$

Thus, with high probability, $\mathcal{D}$ will find some point $x' \in S$ with $||x' - \widehat{s}||_p \leq cr - \Delta$. For that point, we have that:

$$||x' - q||_p \leq ||x' - \widehat{s}||_p + ||\widehat{s} - q||_p \leq cr - \Delta + \Delta = cr$$

Therefore, Algorithm 6 will output a correct answer. If there doesn't exist such a point $x$, it is valid for our algorithm to output $\bot$, so are done.

For the runtime, recall that $|\widehat{S}| \leq O(2Cd^{1/p}/\Delta)^d$. Hence, in order to guarantee success for all queries in $\widehat{S}$, a similar analysis as to the one for the Hamming Hypercube shows that $\mathcal{D}$ takes up:

$$O\left[d \cdot n^{1 + \frac{1}{2c'^2 - 1}} \log\left(\frac{2Cd^{1/p}}{\Delta}\right)\right]$$

space for pre-processing and

$$O\left[n^{\frac{1}{2c'^2 - 1}} \log\left(\frac{2Cd^{1/p}}{\Delta}\right)\right]$$

time per query processed, where

$$c' := \frac{cr - \Delta}{r + \Delta}$$

Note that we use the optimal LSH algorithm for $\ell_p$ spaces, which guarantees $\rho = \frac{1}{2c^2 - 1}$. Our only constraint is that we must have $\Delta < cr$. If we set $\Delta = \frac{c}{10}r$, we get a per-query runtime of:

$$O\left[n^{1 + \frac{1}{2c'^2 - 1}} \log\left(\frac{20Cd^{1/p}}{cr}\right)\right], \quad \text{where } c' = \frac{9c}{10 + c}$$

$\square$

### D.2.3    Removing the dependency on the scale

Our algorithm from Theorem D.5 crucially depends on $\log C$, where $C$ is a bounding box for the query and input point space in the $\ell_p$ norm. We can remove the dependency on $C$ by designing our covering to be data dependent, instead paying an additional logarithmic factor.

Our new covering $\widehat{S}'$ will be a collection of $n$ $\Delta$-coverings, as constructed in Algorithm 6, each one discretizing the $r$-ball around a point $p \in S$. The number of points in this new covering is:

$$|\widehat{S}'| \leq O\left[n \cdot \left(\frac{r \cdot d^{1/p}}{cr}\right)^d\right] = O\left[n \cdot \left(\frac{d^{1/p}}{c}\right)^d\right] \tag{10}$$

Note that the size of this covering improves upon the $(nd)^d$ size of the covering given in [24], which results in a slightly better runtime. This new covering notably does not cover every possible query. However, it covers exactly the queries we care about. This improved covering leads to the following *for-all* guarantee for robust ANN:

**Theorem D.6.** *There exists an adversarially robust algorithm solving the $(c, r)$–ANN problem in the $(\mathbb{R}^d, \ell_p)$ metric space that can answer an unbounded number of adversarial queries. The pre-processing time / space is $\widetilde{O}(nT)$ and the time per query is $\widetilde{O}(T/d)$, where:*

$$T = O\left[d \cdot n^{\rho'} \left(d \log d + \log n\right)\right] \tag{11}$$

*where:*

$$\rho' = \frac{1}{2c'^2 - 1} = \frac{(10 + c)^2}{161c^2 - 20c - 100}$$



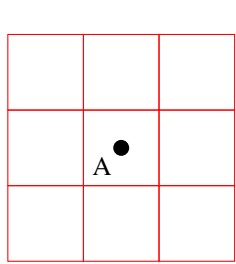

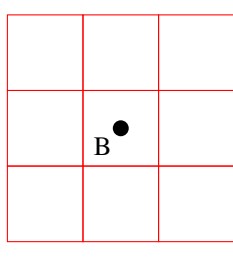

Figure 3: Data-Dependent Discretization of the input query space.

*Proof.* We distinguish between two cases:

1. If a query $q$ is not included in any $B_S(p, r)$ for any $p \in S$, then the answer can safely be $\perp$ because $B_S(q, r) = \emptyset$ necessarily. Thus, we can just run the default LSH algorithm and simply output whatever it outputs.

2. Otherwise, a query $q$ can be included in some $B_S(p, r)$ for some $p \in S$. Then, suppose $\widehat{s'} \in \widehat{S'}$ is a point in our covering such that $||q - \widehat{s'}||_p \leq \Delta$. Then:

$$||p - \widehat{s'}||_p \leq ||p - q||_p + ||\widehat{s'} - q||_p \leq r + \Delta \tag{12}$$

Thus, as we argued before, with high probability $\mathcal{D}$ finds some point $x \in S$ with $||x - \widehat{s'}||_p \leq cr - \Delta$, and for that point we have:

$$||x - q||_p \leq ||x - \widehat{s'}||_p + ||\widehat{s'} - q||_p \leq cr - \Delta + \Delta = cr \tag{13}$$

which means our algorithm will output a correct answer.

As before, our algorithm's space and runtime guarantees scale with $\log |\widehat{S'}|$. $\square$

# E   An Improvement to Exact Fair ANN

A **fair** algorithm outputs, on input $x$, a uniformly distributed output over some pre-determined space of outcomes. In the problem of *exact fair approximate nearest neighbor search*[5], we aim to output a point uniformly in $B_S(q, r)$. Fair ANN algorithms have been studied extensively by Aumüller, Har-Peled, Mahabadi, Pagh, and Silvestri [9]. Their techniques, which inspired the design of Algorithm 3, involved the use of LSH and sampling to yield an fair ANN algorithm whose runtime scales with our familiar $\frac{n(q,cr)}{n(q,r)}$. They prove the following theorem:

**Theorem E.1.** *There exists a fair ANN algorithm using $\widetilde{O}(n^{1+\rho})$ bits of space and $\widetilde{O}(dn^\rho \cdot \frac{n(q,cr)}{n(q,r)})$ time per query, where $\rho$ is an LSH parameter.*

---

[5]Approximate notions of fairness are also studied in [9] and our approach likely extends to those concepts as well. For simplicity in presentation, we focus on the most straightforward definition of fairness.

Our concentric LSH construction can yield an exact fair ANN algorithm with an almost purely sublinear query time, modulo the outlier structural assumption. The algorithm and analysis remain the same, but instead of $\mathcal{L}_{\text{sampling}}$, we apply it to the fair ANN algorithm of Theorem E.1:

**Theorem E.2.** *Given Assumption 1, there exists an algorithm that on query $q \in \mathcal{M}$ outputs a point in $B_S(q, r')$ uniformly at random, where $r' \in [r, cr]$, and $c, r$ are the LSH parameters. If such points do not exist, the algorithm outputs "$\perp$" with high probability. The algorithm uses $O(n^{1+\frac{1}{K}+\rho} \cdot \log n)$ bits of space overall and $O(dn^{\rho+1/K} \log n)$ time per query.*

**Remark (Different Notions of Fairness).** *Our algorithm returns a uniformly sampled point from a sphere with radius $r'$, which is potentially different from $r$. This radius $r'$ depends on $S$, $q$ and the internal randomness used, which makes our guarantee technically different from the one given by Theorem E.1. However, the output is nevertheless fairly produced among a set of valid candidate points.*

**Remark (Fairness and Robustness).** *A natural follow-up question is whether a connection exists between fairness and adversarial robustness. One might intuitively argue that fair algorithms are inherently robust because they don't exhibit bias in their internal randomness toward any specific output. However, this is not always the case. We can always construct an oblivious decider by simply wrapping it around a fair algorithm, and we have seen that deciders are not necessarily robust. Nevertheless, an interesting direction for future work is to quantify levels of robustness and position fair algorithms along this spectrum.*

