# OpenReview forum: "From Search to Decision: A Framework for Adversarially Robust Approximate Nearest Neighbor Search"
_NeurIPS.cc/2025/Workshop/Reliable_ML — NeurIPS 2025 - Reliable ML Workshop_

### Official Review · Reviewer_Z7cx · 2025-09-19
**Answering a Search Query by Robust Deciding for Nearest Neighbour**

**Rating:** 7
**Confidence:** 3

**Review:**

### Summary:
A general reduction from search to a robust decision problem via a binary search tree. An oblivious decider is built using sampling-based LSH where an application of differential privacy yields robustness to adaptive queries. For ANN, a concentric annuli LSH construction improves efficiency except on degenerate datasets, and the paper also gives for-all guarantees (correct on all queries w.h.p.) in low dimensions.
### Strength:
The reduction from search to a robust decision problem is clean and broadly usable, and the write-up makes it easy to follow. The DP-based aggregation handles adaptive queries in a principled way, and the bounds are explicit about the $\sqrt{Q}$ and $+1/K$ terms. The ANN instantiation by sampling-based LSH with concentric annuli gives real speedups under non-degeneracy, while the baseline still covers the general case. I also appreciate the inclusion of for-all results in low dimensions; it makes the overall picture feel both solid and implementable. Additionally, it improves over known fair ANN implementation and they provide a discussion separating this work from a concurrent work.
### Weakness:
Please expand on degenerate datasets: how to spot them in practice, how to guard against them (some pre-processing or mapping?), and why such cases are unlikely in common data distributions. You also mention in the discussion section the goal to derive lower bounds, that would be very interesting to put the results and dependence on $\sqrt{Q}$ and $+1/K$ into perspective.

---

### Official Review · Reviewer_byXe · 2025-09-19
**good topic and nice problem, but want more formalizations**

**Rating:** 6
**Confidence:** 2

**Review:**

This paper develops a framework for adversarially robust Approximate Nearest Neighbor (ANN) search, where an adversary controls both the dataset and adaptive queries. The authors reduce robust search to robust decision problems, applying differential privacy to obtain robustness.

I really like this topic, and think there is something fairly signficant here. However, I found it difficult to

The main weakness for me is clarity. For a main conference submission, I would want the paper to be much clearer in explaining core definitions and intuitions. Some specific points that could be improved:
	1.	Adversarial queries: The paper should define explicitly what an “adversarial query” is and what the adversary’s goal is. Right now, it assumes familiarity with the adaptive adversary model, but readers outside this subcommunity will struggle. Who issues queries, what information they see, and what “breaking” the algorithm means should be made precise.
	2.	Attack of Kapralov et al. [43]: The text says informally that an adaptive adversary can use a polylogarithmic number of queries to “learn enough to force failure.” I would prefer a much more precise restatement of what exactly is learned and how this leads to failure.
	3.	Oblivious decider: This is a central term in the framework but is not defined clearly up front. For non-specialists, it would help to spell out: an oblivious decider is one that succeeds if the query sequence is fixed in advance, not adaptive.
	4.	K-degenerate datasets: The paper defines this formally but does not provide much intuition. Why is this assumption necessary, what are examples of degenerate datasets, and why are they contrived in practice? A worked-out intuition would help.

I do support this work being accepted.